# Assessment of the Equivalence of Low-Cost Sensors with the Reference Method in Measuring PM$_{10}$ Concentration Using Selected Correction Functions

**Tomasz Owczarek** [1,*], **Mariusz Rogulski** [2] **and Piotr O. Czechowski** [1]

1   Faculty of Entrepreneurship and Quality Science, Gdynia Maritime University, Morska 81-87, 81-225 Gdynia, Poland; p.o.czechowski@wpit.umg.edu.pl
2   Hydro and Environmental Engineering, Faculty of Building Services, Warsaw University of Technology, Nowowiejska 20, 00-653 Warsaw, Poland; mariusz.rogulski@pw.edu.pl
*   Correspondence: t.owczarek@wpit.umg.edu.pl; Tel.: +48-502-085-307

**Abstract:** The aim of the work is to demonstrate the possibility of building models to correct the results of measurements of particulate matter PM$_{10}$ concentrations obtained using low-cost devices. Such devices apply the optical method to values comparable with those obtained using the reference gravimetric method. An additional goal is to show that the results corrected in this way can be used to carry out the procedure for testing equivalence of these methods. The study used generalized regression models (GRMs) to construct corrective functions. The constructed models were assessed using the coefficients of determination and the methodology of calculating the measurement uncertainty of the device. Measurement data from the two tested devices and the reference method were used to estimate model parameters. The measurement data were collected on a daily basis from 1 February to 30 June 2018 in Nowy Sącz. Regression allowed building multiple models with various functional forms and very promising statistical properties as well as good ability to describe the variability of reference measurements. These models also had very low values of measurement uncertainty. Of all the models constructed, a linear model using the original PM$_{10}$ concentrations from the tested devices, air humidity, and wind speed was chosen as the most accurate and simplest model. Apart from the coefficient of determination, expanded relative uncertainty served as the measure of quality of the obtained model. Its small value, much lower than 25%, indicates that after correcting the results it is possible to carry out the equivalence testing procedure for the low-cost devices and confirm the equivalence of the tested method with the reference method.

**Keywords:** air pollution; particulate matter; low cost meters; PM$_{10}$ concentrations; equivalence of measurement methods; uncertainty of measurement

## 1. Introduction

A large part of society has been interested in the problem of environmental quality for many years. Water purity, soil and groundwater pollution, waste management, and air quality have become problems that everyone deals with to a greater or lesser extent. This is especially true of air quality which affects everyone and whose negative consequences cannot be alleviated.

Poor air quality can affect people's health, lives, and well-being. Particular attention is paid to particulate matter present in the air. It is a mixture of particles of various sizes, from smaller than 1 m, to larger than 10 m, with different chemical compositions (may contain benzo(a)pyrene, dioxins, furans, or other heavy metals) and from various sources (resulting from natural erosion or human activity). Particulate matter (PM) is most commonly classified by particle size into: PM$_{10}$, PM$_{2.5}$, and

$PM_1$, (i.e., fractions containing particles with a diameter smaller than 10, 2.5, and 1 m). The fraction of the largest particles is currently studied most often and most fully, while the fraction of the smallest particles is still relatively rarely studied due to technical difficulties. However, it is suspected of having the most destructive impact on health, therefore the number of analyses of this group of dusts is increasing [1–4].

Many studies indicate the contribution of airborne pollutants, in particular PM, to the occurrence or exacerbation of symptoms of many diseases, including upper respiratory tract diseases, asthma, chronic obstructive pulmonary disease (COPD), pneumonia, and also cardiovascular disease. The impact of contaminants on the severity of allergy symptoms and general well-being has also been observed. It is recommended for chronically ill people to avoid prolonged exposure to contaminated air. Particularly fine fractions of $PM_{2.5}$ and $PM_1$ dust penetrate deep into the lungs and even into the circulatory system, spreading pollution throughout the body. Research also shows the possibility of transmitting diseases by large dust particles in the air. The presence of viruses, including coronaviruses, has been observed on particulate matter, and the possibility of infection in this way is not excluded [4–10].

Knowledge about air pollution is becoming crucial for many people. Information on the state of air cleanliness is sought. We expect this information to be accurate, current, and available in the area that concerns us. This leads to the need for ever closer monitoring of air quality. The methods of pollution monitoring that have been used for a long time, including reference methods for a given pollution, are effective and considered to be the most accurate. Unfortunately, they are also expensive and require a large amount of time to perform, therefore, monitoring points are located relatively sparsely. The solution to this problem is to use alternative methods of monitoring pollution. Devices introducing alternatives to reference methods of air pollution assessment are being introduced to the market. In the case of solid matter, these methods include the Tapered Element Oscillating Microbalance (TEOM sampler), a method based on the absorption of beta radiation (Eberline sampler), and an optical method involving the analysis of laser light reflected from particles. These methods allow a significant reduction in the size of measuring devices, even down to a portable size, while also reducing the costs of their installation and maintenance (they do not require replacement of filters or laboratory servicing as is the case with the gravimetric method) [1,11]. A significant proportion of these devices also have the ability to send analysis results by electronic means directly to recipients. This allows the installation of measuring devices in places where up till now it has been impossible to monitor air quality. Measuring devices are being installed in streets, schools, public buildings, housing estates, and industrial plants, wherever information about the state of pollution is expected [1,12–14].

However, for the measurements obtained to have a cognitive value, it is necessary to ensure the comparability of results. In particular, it is necessary to ensure consistency between the measurements obtained using the reference method and the measurements from devices using alternative measurement methods. Due to the use of other methods, these devices measure pollution in a different way, which may result in deviations from the reference results. It is therefore necessary to ensure that the differences in measurements obtained by different methods are small and random, and do not contain systematic errors that could disqualify the measurements obtained in the long run. The procedure for testing the equivalence of measurement methods with the reference method for particulate matter was described in the EU Guide to the Demonstration [15]. It envisions carrying out measurements of PM concentrations using the tested device (candidate method, CM) in the immediate vicinity of the device using the reference method, over a sufficiently long period of time and in different weather conditions, and then comparing the series obtained in this way with the help of measurement uncertainty. Any device whose measurements are to be used, and in particular comparable with others, should pass the equivalence test [1,2,12–18].

The measurement results obtained from electronic devices are saved as different values of the output signals from the meter. To determine the concentration of a pollutant in the air, it is necessary to convert it into concentration values. This is done by using calculation procedures (functions) encoded in the device. Such functions may be more or less accurate, depending on the conditions in which

they were created and on the data on the basis of which they were built. Inaccuracies in the operation of the measuring device may not arise from an incorrect measuring method or faulty sensor design but as a result of an imperfect translation function embedded in the device software. In a properly constructed device, it is possible to improve the results by using corrective functions. Such functions can use not only the measurements obtained by the sensors, but also the measurements of features that significantly affect the functioning of the device or the conditions of measurements [16].

The purpose of this study is to demonstrate the possibility of constructing a function correcting the measurements of the low-cost $PM_{10}$ concentration measuring device to values comparable with the measurements derived from the reference method, to prove that the equivalence of these measurement methods can be tested. This function can be used by the manufacturer to change the device software or, if such a change is not possible (even if no other measurements are available), to correct the measurement values at their recipients.

Each of the measuring devices functions in different ways. Consequently, decisions about the need to correct the results obtained and the way this is carried out are made independently for each type of device or even for each model. The obtained correction model is not universal for all $PM_{10}$ electronic meters, but it is only appropriate for the device used in the study. In addition, measurements from electronic sensors are usually corrected by correction factors or simple linear models (using only measurements from electronic devices) [11,19,20]. This leads in many cases to a situation in which the measurement results are correct within a certain range of values of factors affecting the functioning of the device (temperature, concentration of pollutants). Outside this range of values, the measurement results often deviate significantly from the actual values. In this study, the authors want to demonstrate the possibility of using non-linear models and models using weather factors for correction. The correction functions obtained in this way should be more flexible, and the obtained measurement errors less susceptible to changes in the measurement conditions.

## 2. Materials and Methods

The study used measurements of particulate matter ($PM_{10}$) concentrations from low-cost electronic measuring devices using optical sensors. These devices are a new product and are just being launched. Their manufacturer did not allow the device name to be disclosed. They use an optical method for measurements. The device draws a specific amount of air into the reactor, which is illuminated by means of a laser beam of a certain length. Sensors installed in the reactor count the number of light reflections and its parameters and on this basis the concentration of pollutants in the air is calculated. Information on this subject is processed by electronic systems located in the device with a frequency of one measurement per minute. The obtained results are then sent to the recipient via a mobile network. The device allows the measurement of concentrations of various pollutants in the air, including $PM_{10}$, $PM_{2.5}$, and $PM_1$. This also gives the opportunity to take measurements of many weather parameters. The design of the devices ensures their high mobility and measurement of pollutant concentrations virtually anywhere. This device, like many others, has already been described many times, and the results of its measurements have been analyzed in many aspects [10–13,16,17,21].

The study was conducted from 1 February to 30 July 2018 in Nowy Sącz in southern Poland by means of two electronic devices using the optical method of measuring $PM_{10}$ concentrations (candidate method, CM). This period included both cold days in winter and early spring as well as very warm days in summer. This provided a cross-section of the various operating conditions of the devices. The measuring devices were placed in the immediate vicinity of the air pollution measuring station belonging to the Wojewódzki Inspektorat Ochrony Środowiska (Voivodship Inspectorate for Environmental Protection, WIOS), so that they could examine the composition of the same air. The use of two devices using the candidate method is intended not only to better assess equivalence with the reference method, but also to demonstrate the repeatability of measurements carried out by this method [15]. Electronic devices tested $PM_{10}$ concentrations every minute, then these data were aggregated into daily averages. Unreliable measurements were removed from the data series using the

Grubbs test. In this way two series of PM$_{10}$ concentrations were obtained from each of the electronic devices used in the study (CM1 and CM2 expressed in μg/m$^3$). These data were supplemented with daily PM$_{10}$ concentrations from the WIOS station, using the gravimetric method for measurements, as a reference method (RM) [22] and with meteorological data such as wind speed (WV, in m/s), relative humidity (humid, in %), and air temperature (temp, in °C). The data collected in this way were used to assess the equivalence of the test method with the reference method.

Figure 1 shows the PM$_{10}$ concentration values obtained with both tested devices and with the reference method. In the winter months, the concentrations obtained using all the methods are high with a large dispersion of results. In warm months, PM$_{10}$ concentrations clearly decrease. The dispersion of concentration values also decreases.

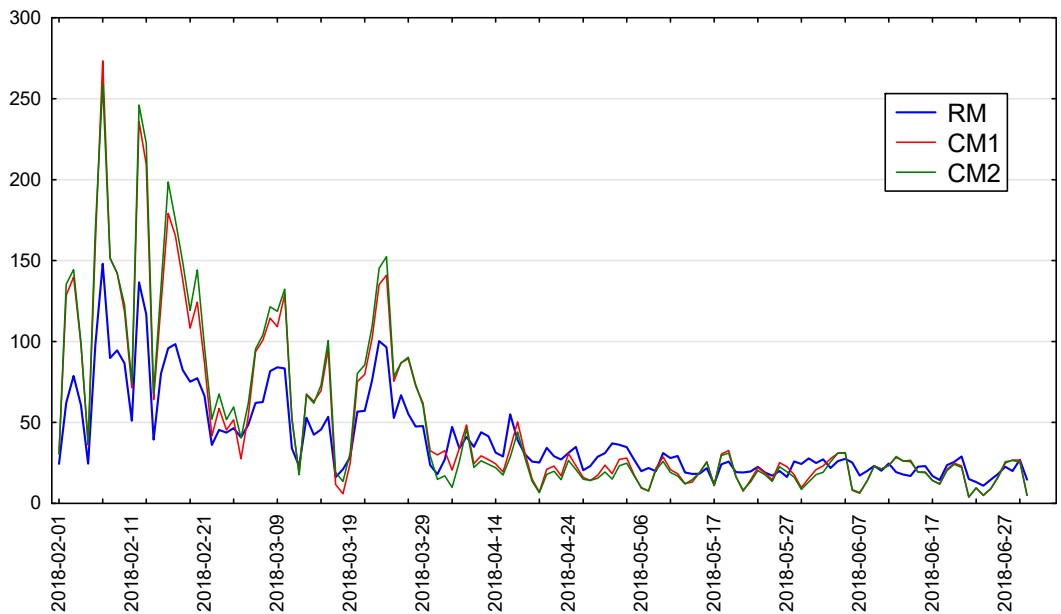

**Figure 1.** Average daily concentrations of PM$_{10}$ from the reference method (RM) and candidate method devices (CM1 and CM2) on 1 February–30 July 2018 in Nowy Sącz.

One can also observe a great similarity in the changes in values obtained from the reference method and from the candidate methods. These values differ quite significantly in the cold months, but while concentration changes, their strength and direction are similar. These observations confirm the values of Pearson's linear correlation coefficients (Table 1).

**Table 1.** Pearson's linear correlation coefficients for PM$_{10}$ concentrations obtained by reference method (RM) and candidate methods (CM1 and CM2).

| Variable | CM1 | CM2 |
|----------|-------|-------|
| RM | 0.968 | 0.965 |

Based on the correlation coefficients, a very large correlation of the concentration values obtained with both candidate devices can be found. It should be emphasized that these values refer to the pure (uncorrected) results obtained from the devices. Both results are statistically significant at any reasonable level of significance.

The values of selected numerical characteristics for concentrations of PM$_{10}$ provided by both candidate devices and by the reference method, as well as the values of the characteristics for the other variables used in the study are presented in Table 2.

**Table 2.** Values of basic numerical characteristics for variables used in the study.

| Variable | Mean | Median | Minimum | Maximum | St. Deviation |
|---|---|---|---|---|---|
| RM | 40.157 | 28.9 | 10.8 | 148.1 | 26.927 |
| CM1 | 49.874 | 26.9 | 3.8 | 273.4 | 50.860 |
| CM2 | 50.872 | 25.5 | 4.0 | 259.2 | 53.709 |
| Temp | 11.697 | 15.3 | −12.6 | 26.0 | 9.771 |
| Wind speed (WV) | 0.913 | 0.8 | 0.3 | 2.7 | 0.355 |
| Humid | 79.531 | 78.4 | 51.5 | 99.9 | 11.136 |

Based on the data from Table 2 and Figure 1, it can be stated that the results obtained by both candidate devices are similar, but they are shifted with respect to each other. This would indicate the need to calibrate the devices after production so that their results have similar values. After correcting the measurement results using the function CM1 = 0.94 × CM2 + 1.9, results with a very high degree of comparability were obtained. The between-sampler uncertainty $u_{BS}$ value was 0.39 µg/m$^3$, where $u_{BS} = \sqrt{\frac{\sum_{i=1}^{n}(CM1_i - CM2_i)^2}{2n}}$. We can accept devices for which the values of this measure are less than 2.5 µg/m$^3$ [15].

There is also a clear shift in the concentration values obtained from the candidate devices compared to the reference method. In addition, much greater variability in the results can be observed. This is especially evident in the cold test period. This indicates the need to correct the results from these devices with the help of a correction function prior to commissioning.

The measurement data were divided randomly (using a systematic random method) into two parts: training data (the part teaching the model) and test data (the part verifying the model). This approach is called cross-validation. In the teaching part there were 90 observations on whose basis the parameters of corrective models were estimated. The second part of the data included 44 observations and they were used to assess the quality of constructed models [23,24].

The generalized regression method (GRM) was used in the construction of corrective function models. It allows the construction of models with different functional forms, using different systems of variables. For selection of the best models in groups (i.e., models with statistically significant parameters and the highest coefficients of determination), backward stepwise regression was used. Calculations were made in the Statistica package [14,25–29].

The models were constructed with PM$_{10}$ concentration from the reference method as the dependent variable, PM$_{10}$ concentration from candidate method (separate modeling was carried out for each of the devices), and the factors that may significantly affect the behavior of the device (i.e., temperature, humidity, and wind speed as independent variables). Due to possible non-linear relationships between independent variables and the dependent variable, models with non-linearly transformed variables were also used. For this purpose, natural logarithm ($ln\ X_i$), exponential transformation ($e^{Xi}$), and second-degree polynomial function were used. In this way eight groups (clusters) of models were obtained which were subject to separate estimation procedures and selection of the best models:

1. linear models,
2. linear models with variable interactions,
3. models based on second-degree polynomials,
4. models based on second-degree polynomials with variable interactions,
5. models using independent variables with their logarithms,
6. models using independent variables along with their logarithms and variable interactions,
7. models using independent variables with their exponential transformations,
8. models using independent variables along with their exponential transformations and variable interactions.

In each group of models, all possible models were estimated, models with significant estimates of structural parameters were selected, and among them the model with the highest value of the adjusted

coefficient of determination $\overline{R}^2$. Then the models obtained in this way were evaluated with one another using the adjusted coefficient of determination based on the training data set.

Later in the study, the functioning of each model was verified using the data verification part. Such a study should verify the occurrence of systematic differences between empirical data and model data and assess the magnitude of random errors. For this purpose, linear regression models (calibration models) were built between the values of the actual concentrations of the reference device and the theoretical values calculated using the previously obtained model in the form of:

$$Y = b_0 + b_1 X \tag{1}$$

The effectiveness of the selected regression model was assessed using the adjusted coefficient of determination and the similarity of the calibration model to the identity function. The calibration model should have an intercept value ($b_0$) not significantly different from 0 and a slope value ($b_1$) not significantly different from 1. Compliance with these requirements leads to the conclusion that the calibration function is an identity and the differences between the variables are in fact random. Both significances were tested using the *t*-value test, while in the second case the significance of the slope value was examined after subtracting one from it. The number of random errors was measured using adjusted $\overline{R}^2$ [15].

The main task of the equivalence test is to determine the uncertainty value of the candidate method. This is associated with the uncertainty of all identifiable factors that may affect the quality of the measurements obtained and is represented by their dispersion (variability). The concept of the calibration function is appropriate to the methodology for testing the equivalence of $PM_{10}$ concentration measurement methods described in the Guide to the Demonstration [15]. These uncertainties are: uncertainty (inaccuracy) of the reference method—$u^2(x_i)$; uncertainty caused by the regression model combining both methods $S_e^2$, caused by the imperfection of estimating the parameters of this model; and the uncertainty of the calibration function (1). *LV* is the maximum allowed concentration of $PM_{10}$ which is 50 µg/m$^3$. Combined uncertainty can be written as:

$$u_{CR}^2 = S_e^2 - u^2(x_i) + [b_0 + (b_1 - 1) * LV]^2 \tag{2}$$

Constructed on their basis combined uncertainty $u^2_{CR}$ is used to calculate the expanded relative uncertainty $W_{CM}$:

$$W_{CM} = k \cdot \sqrt{\frac{u_{CR}^2}{LV}} \tag{3}$$

where *k* is the coefficient of extension calculated on the basis of the Student's *t* distribution with the assumed significance *p*. Most of the computation is assumed to approximate the *k* = 2.

Large values of the expanded relative uncertainty of the candidate method indicate its low usefulness for approximating actual $PM_{10}$ concentrations. Values close to 0 mean that the candidate method gives a satisfactory approximation of the results obtained with the reference method. The limit of acceptance of the method is 25% for extended relative uncertainty [15,30].

However, it should be emphasized that the fulfillment of the condition by expanded relative uncertainty is not the only criterion for recognition of a device as equivalent. The appropriate selection of concentration measurement time and analysis performed for specific groups of measurements are also necessary for this purpose. This part of the analysis is not performed in this study, therefore the results obtained may indicate the possibility of demonstrating the equivalence of methods, but they are not the final proof [15].

The test procedure can be briefly described in a few steps:

(1)   selection of factors affecting $PM_{10}$ concentration measurement in the tested device (i.e., potential independent variables of the correction function);

(2)　selecting all relevant functional forms (groups) of correction functions;

(3)　construction of all models in each group and selection of the best model in the group (i.e., the model that meets all statistical estimation assumptions and gives the highest value of the coefficient of determination);

(4)　evaluation of selected models using a calibration function (i.e., checking for possible systematic errors in the corrected measurements);

(5)　evaluation of selected correction functions employing extended relative uncertainty (assessment of the amount of random errors) and selection of the final form of the correction function.

The correction function indicated as a result of the procedure may be recommended for implementation in the software of the tested device. If the implementation is not possible for technical reasons, the function can be used to correct the value of $PM_{10}$ concentration in the device of the data recipient.

## 3. Results

The following presents the results of an estimate of model parameters in groups. The listings show the best model obtained in stepwise regression. Values of estimates not significantly different from 0 are omitted in all statements.

### 3.1. Linear Model

The linear model of the correction function cluster has the following general form:

$$Y = a_0 + a_1 X_1 + \cdots + a_k X_k \tag{4}$$

where $Y$ is a dependent variable, $X_i$ is an independent variable, and parameters $a_i$ are estimates of structural parameters.

For the group of models formulated above, model parameters were estimated for each combination of independent variables (using the training part of the data), models with insignificant structural parameters were removed, and the best model was selected from the others. Table 3 presents the results of the estimation of the correction model parameters separately for the concentrations from electronic devices CM1 and CM2. The parameter estimates are statistically significant at the significance level of 0.05.

**Table 3.** Estimates of structural parameters for the best models of linear correction function F1 and F2 of the candidate methods $PM_{10}$ concentration, the test statistics *F* value, and the adjusted $\overline{R}^2$.

| Name | Device CM | Intercept | CM | WV | Humid | *F* value | Adjusted $\overline{R}^2$ |
|------|-----------|-----------|------|-------|--------|-----------|---------------------------|
| F1 | CM1 | 39.413 | 0.521 | 3.967 | −0.368 | 806.46 | 0.964 |
| F2 | CM2 | 46.792 | 0.485 | | −0.400 | 1012.31 | 0.958 |

Based on the correction function models described in Table 3, it can be concluded that the corrected concentration of $PM_{10}$ was affected by the concentration obtained from the tested device and the relative humidity of the air. An increase in the humidity value caused a decrease in the corrected $PM_{10}$ value. In the case of air with high humidity, the tested low-cost device tends to overestimate the value of the concentration of pollution. Both models differ primarily in the significance of the influence of wind speed. In the model for the CM1 device, the wind speed is statistically significant, while in the model constructed based on the concentrations from CM2 this variable is statistically insignificant. The difference of significance of the parameter may result from even very small differences in the data. In the case of a parameter on the limit of significance, small differences in the data may cause rejection or non-rejection of the hypothesis about the significance of this parameter in the *t*-test.

The obtained models are properly constructed, which is confirmed by the high values of the *F*-test statistics in the test for the total significance of model parameters (each parameter is also individually statistically significant). The very high values of the coefficient of determination (0.964 and 0.958) in both cases indicate a perfect fit of the model to empirical data. Built models cannot explain 3.6% and 4.2% of the variability of $PM_{10}$ concentrations from the reference device. The remainder of the variation is explained by the model.

The real effectiveness of the indicated models was tested using $PM_{10}$ concentration values from the test part of the data. After calculating the value of the correction functions for these data, their identity with the $PM_{10}$ concentration measurements for the reference method was verified. To this end, calibration functions were built (i.e., linear regression models between corrected concentrations from the tested devices and concentrations from the reference method). The expected result is a linear model for which the slope is statistically insignificantly different from 1, while the intercept is statistically insignificantly different from 0. The estimation of parameters results for calibration function FC1 and FC2 are presented in Table 4.

**Table 4.** Verification of the linear correction function: parameter estimates, estimation errors, adjusted $\overline{R}^2$, and the value of *t*-test statistics for calibration functions F1 and F2.

| Name | Device CM | Slope $a_1$ | Estimation Error $s(a_1)$ | Adjusted $\overline{R}^2$ | $t\ (a_1-1)$ |
|---|---|---|---|---|---|
| FC1 | CM1 | 0.979 | 0.031 | 0.957 | −0.668 |
| FC2 | CM2 | 0.979 | 0.032 | 0.957 | −0.669 |

In both calibration models, the intercept ($a_0$) estimates were significantly indifferent from 0 and for this reason they are not included in Table 4. The slope estimates ($a_1$) are 0.979 for both models and were significantly indifferent from 1. This is confirmed by values $t\ (a_1-1)$ statistics which are close to 0. It can be concluded that both calibration functions are in fact identity transformations, and the concentration values obtained from the correction models do not contain systematic errors.

Very high values of adjusted determination coefficients indicate a high efficiency of the models. In both cases, the calibration models explained the variability of RM concentrations in almost 96% using the values calculated on the basis of corrective models. Only 4.3% of the RM variability were not explained by the model.

Both models of correction functions can be considered very good. The models perfectly match the data from the tested devices for reference measurements, and the results obtained do not contain systematic errors. The corrected concentration values obtained from these models later in this article will be verified for the value of expanded relative uncertainty.

*3.2. Linear Models with Variable Interactions*

The model of the linear correction function group with variable interactions has the following general form:

$$Y = a_0 + a_1 X_1 + \cdots + a_k X_k + a_{k+1} X_1 \times X_2 + \cdots + a_{k+l} X_{k-1} \times X_k \tag{5}$$

where $Y$ is a dependent variable, $X_i$ is an independent variable, $X_i \times X_j$ is the interaction of the *i*-th variable with the *j*-th variable, parameters $a_i$ are estimates of structural parameters, and *l* is the number of all interactions $l = k(k-1)/2$.

Models (5) can include both the independent variables used in the study (i.e., $PM_{10}$ concentrations, temperature, humidity, and wind speed), as well as the interactions of all possible pairs of these variables. The interactions describe the effect of the simultaneous influence of factors on a dependent variable. Table 5 presents the results of the estimation of the correction model.

**Table 5.** Estimates of structural parameters for the best linear models with interactions of the correction function F3 and F4, the value of the *F*-test statistics, and adjusted coefficient of determination $\overline{R}^2$.

| Name | Device | Intercept | CM | Temp | WV | CM × Humid | Temp × Humid | WV × Humid | F Value | Adjusted $\overline{R}^2$ |
|------|--------|-----------|-----|------|------|-----------|--------------|------------|---------|---------------------------|
| F3 | CM1 | 9.087 | 0.880 | | 23.659 | −0.004 | | −0.243 | 620.787 | 0.965 |
| F4 | CM2 | 13.540 | 0.859 | 1.367 | | −0.004 | −0.017 | | 498.528 | 0.957 |

In the F3 model for measurements from the CM1 device, statistically significant factors besides the constant are CM1 concentration, wind speed, interaction of $PM_{10}$ concentration, and humidity, as well as wind speed and humidity. A positive estimate of the structural parameter for wind speed indicates that with stronger winds the $PM_{10}$ concentrations obtained by the CM1 are underestimated. In the F4 model, the CM2 concentration, temperature, interaction of $PM_{10}$ concentration, and humidity as well as temperature and humidity are statistically significant for the CM2. A positive estimate of the structural parameter for temperature indicates that the increase in temperature causes a greater underestimation of the concentration values in the CM2 device. Negative and small values of structural parameters for all interactions indicate that these factors reduce $PM_{10}$ concentration. This impact is relatively small but statistically significant.

The values of the determination coefficients for the F3 and F4 models are close to 0.96 in both cases. This proves that the models are very well adapted to the real $PM_{10}$ concentration values. Both models can be used to correct data from the tested devices.

It should be noted that the F3 and F4 models have significantly different structures for both tested devices. This is undoubtedly a problem when building a universal correction model for all devices of this type. In addition, it can be said that these models are slightly more complex than the linear models F1 and F2. They use more variables and in more complex forms. In return, we do not achieve a significant improvement in the match to real data.

Despite some small reservations about these models, they were subjected to a further assessment procedure using a verification part of the data and a calibration function. The results of the estimation of the parameters of the calibration model are in Table 6.

**Table 6.** Verification of the linear correction function with interactions: parameter estimates, estimation errors, adjusted coefficient of determination values $\overline{R}^2$, and the value of *t*-test statistics for calibration functions FC3 and FC4.

| Name | Device | Slope $a_1$ | Estimation Error $s(a_1)$ | Adjusted $\overline{R}^2$ | $t\,(a_1-1)$ |
|------|--------|-------------|---------------------------|---------------------------|--------------|
| FC3 | CM1 | 0.980 | 0.030 | 0.960 | −0.644 |
| FC4 | CM2 | 0.979 | 0.031 | 0.958 | −0.663 |

As in the previous models, the calibration functions FC3 and FC4 are insignificantly different from the identity. Intercept in both models is equal to 0 while the slope factor is slightly different from 1 (Table 6). The properties of FC3 and FC4 calibration functions indicate that there are no systematic errors in the measurements corrected by the F3 and F4 correction functions.

The match of calculated values based on FC3 and FC4 calibration models to the data from the verification part of the sample is very high. The value of the adjusted coefficients of determination is approximately 0.96. It can be stated that the values calculated on the basis of correction models F3 and F4 are almost perfectly matched to the real concentration values derived from RM. These models can be considered very effective.

It should be noted that the correction functions F3 and F4 fit the electronic device concentration measurements well and ensure their almost perfect compliance with reference concentrations. However, these models are too complicated for practical applications, and it was not possible to build identical correction functions for both devices. Nevertheless, both models will be used in further analysis using expanded relative uncertainty.

### 3.3. Models Using Independent Variables with Their Logarithms

The general form of the correction function cluster containing the logarithms of variables can be written:

$$Y = a_0 + a_1 X_1 + \cdots + a_k X_k + a_{k+1} \ln X_1 + \cdots + a_{2k} \ln X_k \tag{6}$$

where $Y$ is a dependent variable, $X_i$ is an independent variable, $\ln X_i$ is the natural logarithm of the variable, and parameters $a_i$ are estimates of structural parameters.

The group of linear models (5) contains variables describing the factors selected in the study and their logarithms. The use of a logarithm of a variable in the model is aimed at slowing it down, reducing the impact of its changes on changes in the dependent variable. The study used natural logarithms as the ones most commonly used in this type of transformation. Changing the base of the logarithm could increase or decrease the strength of the variable slowdown effect. The results of the estimation of model parameters for the training data set are presented in Table 7.

**Table 7.** Estimate parameters for the best models of the correction function F5 and F6 using variable logarithms, the value of *F*-test statistics, and the adjusted coefficient of determination $\overline{R}^2$.

| Name | Device | Intercept | CM | ln (WV) | ln (Humid) | *F* Value | Adjusted $\overline{R}^2$ |
|------|--------|-----------|------|---------|------------|-----------|------------------|
| F5 | CM1 | 136.661 | 0.528 | 4.975 | −28.055 | 823.406 | 0.965 |
| F6 | CM2 | 151.775 | 0.487 | | −31.352 | 1023.823 | 0.958 |

The F5 and F6 models resulting from the estimation and selection of the best model have a similar structure. In both models, significant factors affecting the dependent variable are the measurement values of $PM_{10}$ concentration from the tested devices and the relative humidity of the air. Negative signs of the structural coefficients of models for humidity (−28.055 and −31.352) indicate, just as in the case of F1 and F2 linear models, overestimation of $PM_{10}$ concentrations in the tested devices caused by high humidity. In addition, the influence of logarithm of wind speed in the F5 model was significant. A positive sign of the structural coefficient shows that the increase in wind speed affects the reduction of $PM_{10}$ concentrations from the tested device in relation to the RM concentrations.

As in the previous groups, the best models in the cluster of models containing variable logarithms can be considered very promising. These models successfully passed the test for the total significance of model parameters (*F*-test). Both models F5 and F6 have very high values of determination coefficients (0.965 and 9.58). This means that both models explain the variability of the dependent variable very well, or convert the measurements from the tested devices into measurements from RM very well.

The further part of the study verified the performance of the properties of the proposed corrective models F5 and F6. The values of structural parameters and values of coefficients of determination for the calibration functions FC5 and FC6 are presented in Table 8.

**Table 8.** Verification of the linear correction function with variable logarithms: parameter estimates, estimation errors, adjusted coefficient of determination values $\overline{R}^2$, and the value of *t*-test statistics for calibration functions FC5 and FC6.

| Name | Device | Slope $a_1$ | Estimation Error $s(a_1)$ | Adjusted $\overline{R}^2$ | $t\,(a_1{-}1)$ |
|------|--------|-------------|---------------------------|------------------|-----------|
| FC5 | CM1 | 0.979 | 0.032 | 0.957 | −0.670 |
| FC6 | CM2 | 0.979 | 0.032 | 0.957 | −0.675 |

Both FC5 and FC6 calibration functions are identity functions in that the slope values are statistically insignificantly different from 1 (which is confirmed by small values of $t(a_1{-}1)$ statistics), while intercept values are insignificantly different from 0. It can be stated that $PM_{10}$ concentration values obtained as a result of the operation of the correction functions F5 and F6 do not contain systematic errors compared to the values derived from RM, and the differences between them are random.

The values of the coefficients of determination are very high and amount to 0.957. This means that the data obtained by both methods (RM and CM after correction) are very well matched and suggests that any errors are relatively small.

It should be considered that the correction functions F5 and F6 meet the assumptions and can be further assessed using expanded relative uncertainty.

### 3.4. Models Using Independent Variables with Their Exponential Transformations

The cluster of models of the correction function containing the exponential transformation in $e^x$ form has the following general form:

$$Y = a_0 + a_1 X_1 + \cdots + a_k X_k + a_{k+1} e^{X_1} + \cdots + a_{2k} e^{X_k} \tag{7}$$

where $Y$ is a dependent variable, $X_i$ is an independent variable, $e^{Xi}$ is the value of the exponential transformation (exponent) of the variable, and parameters $a_i$ are estimates of structural parameters.

These models use the values of the variables selected in the study and their values after applying the exponential transformation. The use of exponential transformation is intended to accelerate the value of a variable. In other words, it causes a greater impact on the changes in the dependent variable than the linear impact of changes in this variable.

The values of the estimated structural parameters of the F7 and F8 correction models for both tested devices are shown in Table 9.

**Table 9.** Estimate parameters for the best models of the correction function F7 and F8 using the exponential transformations of variables, the value of *F*-test statistics, and the adjusted coefficient of determination $\overline{R}^2$.

| Name | Device | CM | Temp | WV | Humid | *Exp* (WV) | *F* Value | Adjusted $\overline{R}^2$ |
|------|--------|-------|-------|--------|--------|--------|---------|------|
| F7 | CM1 | 0.579 | 0.262 | 22.715 | −0.080 | −2.526 | 999.366 | 0.982 |
| F8 | CM2 | 0.565 | 0.414 | 21.589 | −0.093 | −2.392 | 908.694 | 0.981 |

The best models of correction functions F7 and F8 containing exponential transformations have a rather complicated structure. They contain $PM_{10}$ concentration values from the tested devices, air temperature, and wind speed with positive marks of structural coefficient estimates as well as air humidity and exponent from wind speed with negative signs. This means that in this model the increase in temperature and wind speed skews down the results obtained in relation to reference measurements, while the increase in humidity increases it slightly. It should be noted that both constructed models have a very similar structure.

The obtained models have the correct structure, which confirms the positive result of the *F*-test (high values of the *F* statistics). Models F7 and F8 have very high values of coefficients of determination (0.982 and 0.981, respectively). Using them as a corrective function of measuring $PM_{10}$ concentrations from the tested devices gives a very good fit to the data derived from RM. The values of the determination coefficients indicate the greatest ability of these models to correct concentration measurements.

In order to confirm the properties of the F7 and F8 correction functions, FC7 and FC8 calibration functions were built between RM measurements and the values were calculated using the correction functions. Structural parameters of the model and coefficients of determination are shown in Table 10.

**Table 10.** Verification of the correction function with exponential transformations: parameter estimates, estimation errors, adjusted coefficient of determination values $\overline{R}^2$, and the value of *t*-test statistics for calibration functions FC7 and FC8.

| Name | Device | Slope $a_1$ | Estimation Error $s(a_1)$ | Adjusted $\overline{R}^2$ | $t\,(a_1-1)$ |
|------|--------|-------------|---------------------------|---------------------------|--------------|
| FC7 | CM1 | 0.969 | 0.038 | 0.937 | −0.818 |
| FC8 | CM2 | 0.967 | 0.039 | 0.933 | −0.841 |

The values of the estimated coefficients of the calibration function show that it has the expected form. The slope value is insignificantly different from 1 and the intercept value is statistically indifferent from 0. Both calibration functions FC7 and FC8 can be considered as an identity function. The values of the coefficients of determination are very high and amount to 0.937 and 0.933, respectively. Calibration models match very well with RM measurements calculated with the help of a correction function using values from the tested devices. On this basis, as well as in the case of the F7 and F8 correction functions, it can be stated that they do not generate systematic errors and the random errors are relatively small.

In addition, the F7 and 8 models can be used to assess the equivalence of methods with the help of expanded relative uncertainty.

### 3.5. Other Models

The estimation of parameters in the remaining groups of functions did not produce any new significant forms of correction functions. The best models of the individual groups overlap with the models described earlier.

In the model based on the second-degree polynomial, after estimating the structural parameters of the model, it turned out that this model does not differ in structure from the model based on a linear function. In this model, the corrected value of $PM_{10}$ measurements is influenced by the original measurements coming from the tested device, air humidity, and wind speed. Estimates of structural parameters and the value of the coefficient of determination for this model are identical to those for the linear model.

A similar situation occurs in the case of the second-degree response surface model (i.e., the model in which there are variables and all products of variable pairs). This model is a more general model than the one mentioned above. Furthermore, in this case, after estimating the model parameters, it turned out that the best model obtained uses the same variables and parameter estimates have the same values as for the model estimated for the linear function. On this basis, it can be assumed that the $PM_{10}$ concentration values are not related to the squares of any of the variables.

The estimation of parameters for the models and selection of the best model from among all models using variable logarithms and their interactions led to a model in which the corrected $PM_{10}$ concentration was affected only by the original $PM_{10}$ concentration values and natural logarithm from relative humidity. Estimates of structural parameters lead to a model identical to the model built solely on variable logarithms. Therefore, it should be recognized that the interaction of variable logarithms does not have a significant impact on the correction function.

Similarly, in the group of models using the exponential transformations of variables and their interactions, the model that used all the statistical assumptions and at the same time gave the highest value of the coefficient of determination was the model using only exponential transformations. The interaction of variable exhibitors turned out to be statistically insignificant and did not affect the construction of the corrective function.

The study also verified a group of models based on third variable powers (third-degree polynomial), third-degree response surface models (a model in which the products of three variables exist), and a group of models in which variables are delayed by one day. In none of these groups did the built models have satisfactory properties nor did estimation lead to models described earlier. Therefore, in

the construction of the correction function, the functions belonging to these groups of functions should be abandoned.

The groups of models listed in this part of the study led to either the models described earlier or the models with unsatisfactory properties. For this reason, these models will not be further analyzed and discussed.

### 3.6. Measurement Uncertainty of Built-Up Correction Functions

In previous parts of the study, the statistical properties of the best correction models in four groups of models were verified positively. In addition, they all had very high determination coefficient values. The ultimate goal of transforming measurement data using correction functions is to obtain such values that allow demonstrating the equivalence of the reference method with the method used in the devices tested. Therefore, it is necessary to carry out measurement evaluation procedures analogous to those proposed in the "Guide to Demonstration" [15]. This consists of two elements: the construction and possible use of the calibration function and the calculation of extended relative uncertainty.

For all the analyzed models of F1–F8 correction functions, calibration functions were previously built. All these functions had very satisfactory properties. Therefore, it should be recognized that the results obtained employing the correction functions do not require calibration.

The second element of equivalence testing is the calculation of measurement uncertainty. The concept of combined uncertainty (2) and extended relative uncertainty (3) is used for this purpose. The uncertainty value allows assessing the differences between the results obtained with both methods. Extended relative uncertainty can also be understood as a chance to obtain an incorrect estimation of the concentration of $PM_{10}$ by the tested method. Low values of both uncertainty measures are considered better. In the case of expanded relative uncertainty, the method acceptance limit, the method equivalence limit, is 25%. To calculate both measurement uncertainties, the entire available data set (i.e., teaching part of data and verification part of data), were used. The results are presented in Table 11.

**Table 11.** Combined uncertainty and extended relative uncertainty for all best models of correction function.

| Measure | Symbol | Model | | | | | | | |
|---|---|---|---|---|---|---|---|---|---|
| | | F1 | F2 | F3 | F4 | F5 | F6 | F7 | F8 |
| Combined uncertainty | $u^2_{CR}$ | 11.137 | 11.243 | 11.771 | 14.087 | 10.856 | 11.318 | 18.490 | 23.564 |
| Extended relative uncertainty | $W_{CR}$ | 13.3% | 13.4% | 13.7% | 15.0% | 13.2% | 13.5% | 17.2% | 19.4% |

Combined uncertainty values for all corrective functions tested are low. Unfortunately, we cannot indicate an acceptability threshold for this measure. Its assessment is subjective and results solely from experience in testing equivalence of methods. However, it can be seen that the F7 and F8 models using exponential transformations have uncertainty values higher than the others. This may indicate a worse fit of these two models and generation of larger errors when comparing the concentration values.

More explicit results can be obtained by using extended relative uncertainty in the analysis. For all the selected correction functions, the values of extended relative uncertainty are low and the differences between them are relatively small. All values range from 13.2% to 19.4%. It follows that the application of each of the correction functions would lead to compliance with the requirement imposed on expanded uncertainty (i.e., an uncertainty value of less than 25%). It can be assumed that after applying these correction functions, the method should pass the test of equivalence with the reference method.

However, the purpose of this study was to choose the model that is best suited to correct the concentration results obtained by the tested devices. In this case, due to the lowest relative uncertainty values, the F1 and F2 functions built on the basis of linear models and F5 and F6, using models containing variable logarithms, should be indicated. Among these two groups of models, we can indicate the model that will be used to correct measurements of $PM_{10}$ concentrations obtained from

the tested devices. These models, however, differ in terms of the degree of complication (the remaining parameters of model evaluation are at a similar level). Therefore, the authors would be inclined to indicate the linear model F1 and F2 as the best for correcting data from the tested device.

## 4. Discussion

The aim of the study was to demonstrate the possibility of constructing a corrective function for the tested low-cost devices measuring concentrations of $PM_{10}$ particulate matter to values comparable with measurements obtained using the reference gravimetric method. A further aim was to demonstrate the equivalence of both methods. For this purpose, groups of functions (functional forms) were selected that could be used to construct the appropriate corrective function. For each of them, all models were built that can be constructed for a selected set of variables. From among them, the best model was selected in each group (i.e., one that gave the largest value of the adjusted coefficient of determination $\overline{R}^2$ when all stochastic assumptions were met for linear econometric models).

The analysis found that four groups were obtained in models with promising properties and significantly different in terms of design. These include linear models, linear models with variable interactions, linear models with variable logarithms, and linear models with exponential transformations of variables. In the remaining groups, unsatisfactory forms of functions were obtained: with low statistical properties, a very complicated structure or built analogously to the previously constructed models. Eight models of the correction function were obtained in this way, four for each of the tested devices.

The analysis of statistical significance of variables and the values of estimates of their structural coefficients indicate that the measurements of $PM_{10}$ concentrations performed by the tested devices were influenced by humidity and wind force. The air temperature had a very limited influence, although it seemed that it was the factor that should have the strongest effect. In the case of wind force, structural parameters were positive in all models. This means that strong wind causes a decrease in the observed concentration of $PM_{10}$ and the model must later correct this value upwards. In the case of humidity, the reverse is true. Negative values of structural parameters indicate that the increase in humidity level contributes to the apparent increase in $PM_{10}$ concentrations detected by the device.

Comparing the selected models in terms of their ability to approximate $PM_{10}$ concentration values produced by the reference method (determination coefficients) and in terms of the absence of systematic errors, it can be concluded that all models have similar properties in this respect. The values of the adjusted coefficients of determination calculated for both the data training set and the verification set are similar and very high. The values of adjusted $\overline{R}^2$ exceed 0.95. This means that the models are very well adapted to the empirical data (i.e., $PM_{10}$ concentrations from the reference method). All models also passed the systematic failure assessment. Calibration models for all selected correction functions have satisfactory properties, in that they do not differ statistically from the identity function. It can be considered that in terms of design all correction functions meet the assumptions and have similar ability to correctly adjust raw results.

The selected correction functions differed slightly in terms of measurement uncertainty. The values of extended relative uncertainty for all corrective functions were low and met the requirements for equivalent methods. The lowest, and hence the best, uncertainty values were obtained for linear models (F1 and F2) and linear models with variable logarithms (F5 and F6). The values of expanded relative uncertainty for these functions ranged from 13.2–13.5%. These two types of models can be recommended for the correction of data from the tested electronic devices.

The authors of the study, however, were inclined to indicate one type of model for possible implementation in the devices. Due to the smaller complexity of model construction, they chose the linear model for further use. The form of this model can be written as follows:

$$\hat{RM} = 39.413 + 0.521 \cdot CM1 + 3.967 \cdot WV - 0.368 \cdot Humid \tag{8}$$

for device 1, and

$$\hat{RM} = 46.792 + 0.485 \cdot CM2 - 0.400 \cdot Humid \tag{9}$$

for device 2. These models should correct the measurement results from the low-cost devices tested to the values obtained by the reference method. A separate matter is the primary, precise calibration of manufactured devices and the construction of a uniform model for all devices from the same manufacturer.

Guidelines for the verification of equivalence of measurement methods indicate the need for further validation of the method. After demonstrating equivalence, the procedure should sometimes be repeated for different climatic conditions and in different geographical locations [15]. Similarly, this should also be the case with the corrective models proposed in this paper. New data may allow better adjustment of models to real concentration values. We can reduce model parameter estimation errors and minimize the impact of erroneous or outlier observations. Measurement data from other locations can also allow us to build models with better functional forms. It is not impossible to build different models for different types of climatic and geographical conditions or models reacting differently to different values of factors affecting the obtained measurements, for example, models with different parameters for high or low PM concentrations. However, to construct them, additional measurement data from devices conducting real field measurements is needed.

## 5. Conclusions

The study analyzes the functioning of a low-cost electronic device for measuring $PM_{10}$ concentrations in air. The purpose of the work is to demonstrate the possibility of building a correction function for the measurements from the tested device and to prove that the corrected data will provide the opportunity to perform testing for equivalence with the reference method. Several models meeting the requirements were constructed in the study. The best of them was a linear model (8) and (9), using $PM_{10}$ concentration values from the tested devices, wind speed and humidity. It approximated the reference method concentration values almost perfectly. The analysis using expanded relative uncertainty has shown that there is a good chance that after applying the correction it will be possible to demonstrate equivalence with the reference method. This will allow measurements from this device to be treated as equivalent to reference measurements. The obtained model can be used by the device manufacturer to improve the device's functioning.

**Author Contributions:** Conceptualization, M.R. and T.O.; methodology, T.O.; validation, P.O.C.; formal analysis, T.O.; investigation, T.O. and M.R.; resources, M.R.; data curation, M.R. and T.O.; writing—original draft preparation, T.O. and P.O.C.; writing—review and editing, T.O.; visualization, T.O.; supervision, P.O.C. All authors have read and agreed to the published version of the manuscript.

**Funding:** This research received no external funding.

**Conflicts of Interest:** The authors declare no conflict of interest.

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
