# Peer review of "Assessment of the Equivalence of Low-Cost Sensors with the Reference Method in Measuring PM10 Concentration Using Selected Correction Functions"

_sustainability, doi:10.3390/su12135368_

Round 1

Reviewer 1 Report

The paper entitled: “Assessment of the Equivalence of Low-Cost Sensors with the Reference Method in Measuring PM10 Concentration Using Selected Correction Functions” deals with a very interesting topic and faces the problem of the equivalence of the low cost particulate matter samples with respect to traditional methods. Due to the ever increasing diffusion of this type of low cost sensors it really represents an interesting study. Nevertheless in many points the text is quite vague and more details should be given.

Some comments suggestions are reported below while other comments are included in the revised version of the paper. English should be revised in many points.

In the material and methods:

At line 106 please give more details on the measurements (which sensor?)

 I suggest at line 108 to include also this reference:

Fermo, P., Comite, V., Falciola, L., Guglielmi, V., Miani, A.

Efficiency of an air cleaner device in reducing aerosol particulate matter (PM) in indoor environments (2020) International Journal of Environmental Research and Public Health, 17 (1)

At line 117 explain what do you mean with candidate method 1 and 2 since here it is first time they are mentioned. Were obtained with two different low cost sensors? Which kind of sensor?

It seems that the discussion is not complete.

The conclusions are missing at all.

Author Response

Response to Reviewer 1 Comments

Thank you for the comments contained in the review. I think they helped me improve the article test. I tried to refer to all comments in the text of the article.

The article was also improved in terms of language. I hope that in this respect it will already meet the requirements.

  1. At line 106 please give more details on the measurements (which sensor?)
    Response: I expanded the text with more detailed information on the design and functioning of the tested PM10 concentration meter. Information about this can be found in the lines 119-129.
  2. I suggest at line 108 to include also this reference: Fermo, P., Comite, V., Falciola, L., Guglielmi, V., Miani, A. Efficiency of an air cleaner device..

Response: Indicated article was very interesting. I used it in the text. Line 131. Thank you for paying attention to this article.

  1. At line 117 explain what do you mean with candidate method 1 and 2 since here it is first time they are mentioned. Were obtained with two different low cost sensors? Which kind of sensor?
    Response: The study took part in two identical devices using optical methods. The candidate method CM in this case is the device type, and CM1 and CM2 were series of measurements from these two devices. I tried to explain it in the following lines: 133-134, 138-141 and 142-144.
  2. It seems that the discussion is not complete.
    Response: In the part entitled Discussion, my goal wasn’t to develop descriptions regarding the estimated models. To reduce the article text I decided to limit the discussion in this part. Now I consider this approach to be wrong. Therefore, the entire Discussion chapter has been enriched with a discussion of the obtained results. The added or changed text is marked in the chapter. There are a lot of changes, that's why I don't give lines containing them.
  3. The conclusions are missing at all.
    Response: The purpose of the work was to build a correction model for the tested device and to demonstrate that this model can enable testing of equivalence with the reference method. It seemed to me that achieving this goal was correctly described in the Discussion chapter. This chapter has been extended by two additional paragraphs: 550-558 and 583-593. The text also includes a new short chapter summarizing the study (5. Conclusions l. 594-605). The publisher says that this chapter is not necessary, but according to the reviewer's suggestion, we found it useful.
  4. Comments in the pdf file.

Response: We have also corrected the errors indicated in the pdf file. Language errors have been corrected. References were added in the lines 69, 182, the test procedure was re-described – lines 244-253 and expanded the subchapter: Other models – lines 454-488.

Yours faithfully

Authors

Reviewer 2 Report

The paper investigates the possibility of constructing a function correcting the measurements of the low-cost PM10 concentration measuring device to comparable values with the measurements derived from the reference method.

The paper is well structured. the conclusions are clear and supported by results. Despite that, it doesn’t present advancement in the research field. Please, the Authors must explain better the novelty of their research. The authors have to check typos error (ie line 44).

Author Response

Response to Reviewer 2 Comments

Thank you for the comments contained in the review. I think they helped me improve the article test. I tried to refer to all comments in the text of the article.

The article was also improved in terms of language. I hope that in this respect it will already meet the requirements.

Point 1: The paper is well structured. the conclusions are clear and supported by results. Despite that, it doesn’t present advancement in the research field. Please, the Authors must explain better the novelty of their research.

Response 1: The correction functions for PM10 measurements are not universal. Each device type and model must be corrected in a different way. This means that the function obtained in the study is unique and has never been shown before. In addition, correction functions usually have a simple form. In this study, were used nonlinear functions and models using additional factors such as humidity or temperature. The authors provide explanations on this subject in lines 105-116.

Yours faithfully

Authors

Reviewer 3 Report

This article is about the assessment of the low cost sensors with the reference method in measuring PM10. I have some detailed comments listed below to help improving this manuscript. 

1, Line 25, is there any quantification of the correcting results? Need to list here to confirm the results are really reliable to use. 

2, The limitation of study should be elaborated. What's the next step to improve the method. What's the implication of this method? 

Author Response

Response to Reviewer 3 Comments

Thank you for the comments contained in the review. I think they helped me improve the article test. I tried to refer to all comments in the text of the article.

The article was also improved in terms of language. I hope that in this respect it will already meet the requirements.

Point 1: Line 25, is there any quantification of the correcting results? Need to list here to confirm the results are really reliable to use. 

Response 1: The only assessment of the correct functioning of the PM10 measuring device and the function to correct its measurements is proof of equivalence with the reference method. In our study, it is not possible to carry out such proof, but we use significant elements of the equivalence test to assess corrective functions. These are calibration functions (and their properties) and assessment of measurement uncertainty. We wrote about it in lines 25-28.

Point 2, The limitation of study should be elaborated. What's the next step to improve the method. What's the implication of this method? 

Response 2: I am analyzing the functioning of this device, actually analyzing its measurements, for some time. I intend to continue this work. However, new measurements and tests in other locations are necessary for this. By using them it will be possible to improve the models and, above all, to be sure of their versatile use. However, this requires several years of data collection (different places and different weather conditions). Of course, the model will be handed over to the manufacturer for use in the device. We wrote about this in lines 583-593.

Yours faithfully

Authors

Round 2

Reviewer 1 Report

The authors have introduced the modifications required and in my opinion the paper could  now be published in the present form.

Reviewer 3 Report

this version if good.